# Parafermions in moiré minibands

Hui Liu ✉, Raul Perea-Causin ✉ & Emil J. Bergholtz ✉

Moiré materials provide a remarkably tunable platform for topological and strongly correlated quantum phases of matter. Very recently, the first Abelian fractional Chern insulators (FCIs) at zero magnetic field have been experimentally demonstrated, and it has been theoretically predicted that non-Abelian states with Majorana fermion excitations may be realized in the nearly dispersionless minibands of these systems. Here, we provide telltale evidence based on many-body exact diagonalization for the even more exotic possibility of moiré-based non-Abelian FCIs exhibiting Fibonacci parafermion excitations. In particular, we obtain low-energy quantum numbers, spectral flow, many-body Chern numbers, and entanglement spectra consistent with the $\mathbb{Z}_3$ Read–Rezayi parafermion phase in an exemplary moiré system with tunable quantum geometry. Our results hint towards the robustness of moiré-based parafermions and encourage the pursuit in moiré systems of these non-Abelian quasiparticles that are superior candidates for topological quantum computing.

In the last years, moiré materials have been established as an accessible and highly tunable platform for exploring strongly correlated topological phases of matter[1–7]. Most notably, specific van der Waals heterostructures consisting of twisted layers of graphene or transition metal dichalcogenides have been theoretically predicted[8–15] and experimentally observed[16–22] to host fractional Chern insulators (FCIs) —lattice analogs of fractional quantum Hall (FQH) states that can exist in the absence of a magnetic field[23–33]. So far, the FCI states that have been unambiguously identified in experiments are topologically equivalent to Abelian hierachical FQH states[34–37] hosting Abelian anyon excitations[38,39]. While the possibility of exploring anyon physics at zero magnetic field and moderate temperatures is already remarkable, the pursuit of non-Abelian quasiparticles—which should appear as elementary excitations in more exotic FCI states—constitutes a more ambitious and potentially rewarding venture with prospects for fault-tolerant topological quantum computation[40].

Recent experimental signatures[41], suggested to arise from non-Abelian topological order at moiré filling fractions with even denominator, have sparked a series of theoretical works addressing the stability of Moore–Read (MR) states in moiré systems[42–49]. Strengthening the theoretical evidence based on the ground state degeneracy, which can correspond to either the MR phase or charge density waves (CDWs), further analysis counting quasi-hole excitations in entanglement spectra has shown that MR states could indeed be stable in moiré materials[42,47,48]. Unfortunately, the braiding operations allowed by the emergent Majorana excitations in MR states cannot generate any arbitrary unitary transformation and would thus be insufficient for universal quantum computation[40]. This issue can, however, be overcome by exploiting parafermions with richer braiding statistics, which have been predicted to appear in FQH systems[50], FQH–superconductor heterostructures[51,52], and certain non-Hermitian systems[53], although their experimental realization remains elusive. A natural step forward in the context of moiré FCIs is thus to search for Fibonacci parafermions appearing in the $\mathbb{Z}_3$ Read–Rezayi (RR) state[50]— which is characterized by the clustering of composite fermions into triplets, as opposed to the pairing in the MR phase. Realizing parafermion excitations in the widely accessible and tunable moiré materials would open an exceptionally promising avenue in the development of fault-tolerant topological quantum computers.

In this work, we provide numerical evidence showing that the non-Abelian RR FCI at 3/5 filling can be realized in a moiré system. In particular, we perform many-body exact diagonalization on an exemplary moiré flat band based on a double twisted bilayer graphene (dTBG) model with tunable quantum geometry. First, we obtain the expected number of degenerate ground states appearing at momenta fulfilling the generalized Haldane statistics. After that, we verify the persistence of the many-body energy gap upon insertion of magnetic flux (i.e. twisted boundary conditions) and find that, on average, each ground

Department of Physics, Stockholm University, AlbaNova University Center, Stockholm, Sweden. ✉e-mail: hui.liu@fysik.su.se; raul.perea.causin@fysik.su.se; emil.bergholtz@fysik.su.se

state has a many-body Chern number of 3/5. More strikingly, we show that the state counting in the low-energy sector of the particle-cut entanglement spectrum (PES) exactly matches the number of allowed quasi-hole excitations. In addition, we find that the RR phase is destroyed when the average quantum metric of the considered flat band approaches asymptotically the ideal value for the second Landau level (LL), highlighting both the usefulness and the limitations of this quantity as a heuristic indicator for the emergence of non-Abelian phases. Finally, we show that the RR phase is less stable at 2/5 filling but still present—as reflected in the quasi-particle excitations accessed through the hole entanglement spectrum—, signaling the robustness of moiré-based parafermion FCIs.

## Results
### Setup
We consider an exemplary model based on dTBG describing moiré minibands with tunable quantum geometry. Concretely, the single-particle valley- and spin-polarized Hamiltonian reads[54]

$$H_0(\mathbf{r}) = \begin{pmatrix} u_1 I & D^\dagger(\mathbf{r}) & 0 & 0 \\ D(\mathbf{r}) & u_2 I & \gamma I & 0 \\ 0 & \gamma I & u_3 I & D^\dagger(\mathbf{r}) \\ 0 & 0 & D(\mathbf{r}) & u_4 I \end{pmatrix}, \quad (1)$$

where the $2 \times 2$ matrix $D_{11}(\mathbf{r}) = D_{22}(\mathbf{r}) = -2ik_\theta^{-1}\bar{\partial}$, $D_{12}(\mathbf{r}) = D_{21}(-\mathbf{r}) = \alpha U(\mathbf{r})$ describes TBG in the chiral limit[55], $\gamma$ is the coupling between the two TBG sheets, and $u_i$ is a layer/sublattice potential. Here, $k_\theta = 4\pi/3a_m$ ($a_m$ is the moiré lattice constant), $\bar{\partial} = \frac{1}{2}(\partial_x + i\partial_y)$, and $U(\mathbf{r}) = e^{-i\mathbf{q}_1 \cdot \mathbf{r}} + e^{i\phi}e^{-i\mathbf{q}_2 \cdot \mathbf{r}} + e^{-i\phi}e^{-i\mathbf{q}_3 \cdot \mathbf{r}}$ with $\phi = 2\pi/3$ and $\mathbf{q}_n = C_3^n \mathbf{k}_\theta$. Throughout this work, we consider the first magic angle[55], i.e. $\alpha \approx 0.586$, and set $u_1 = 0.1$, $u_2 = u_3 = u_4 = 0$ in order to obtain an isolated flat band.

In this setting, the system is characterized by two consecutive non-degenerate and nearly flat bands exhibiting equal Chern number $\mathcal{C} = 1$ and becoming degenerate in the limit $\gamma \to \infty$ (see Supplementary

Note 1 and Supplementary Fig. 1). The upper band maintains an ideal quantum geometry[56] $\mathrm{tr}[g(\mathbf{k})] = |\Omega(\mathbf{k})|$ as $\gamma$ varies, where $g(\mathbf{k})$ and $\Omega(\mathbf{k})$ are the Fubini–Study (FS) metric and the Berry curvature, respectively. A definition of the FS metric is provided in the Methods section. We focus on the lower band, which has a tunable FS metric with the average $\chi = \frac{1}{2\pi} \int_{\mathrm{BZ}} d^2\mathbf{k}\, \mathrm{tr}[g(\mathbf{k})]$ ranging from $\chi = 1$ as $\gamma \to 0$ to $\chi = 3$ as $\gamma \to \infty$ corresponding to the average quantum metric of the lowest and the first excited (second) LLs, respectively[57]. We note, however, that this correspondence generally demands additional conditions related to vortexability[54]. Motivated by the fact that the RR states are energetically more competitive in the second LL of FQH systems[50,58,59], we employ $\gamma$ as a tuning knob to explore the emergence of this non-Abelian phase in our model.

### Read–Rezayi parafermion states
We start by considering a fractionally filled dTBG with $\nu = 3/5$ and a relatively large coupling $\gamma = 3.75$ between the two TBG sheets. Performing exact diagonalization on systems with $N_s = 20$ and $N_s = 25$ moiré sites (see Methods), we obtain 10 nearly-degenerate ground states separated from the excited states by an energy gap, cf. red dots in Fig. 1a, d. The splitting of these states is generically expected on the relatively small finite size systems accessible to exact diagonalization, but should go away in the large system limit. Moreover, we find that the ground states do not flow into excited states upon flux insertion (i.e. twisted boundary conditions), confirming the presence of a many-body gap, cf. Fig. 1b, e. The 10-fold ground state degeneracy is in agreement with that expected in the thin-torus limit, where a quantum Hall system is adiabatically mapped into a 1D lattice that can be exactly solved[60]. In particular, the RR ground states at filling $\nu = k/(kM + 2)$ are formed by $k$ particles occupying $kM + 2$ consecutive sites and every two particles being separated by at least $M$ sites[50,61,62]. Note that the 1D lattice sites in the thin torus are analogous to momentum points[23]. For fermionic Read–Rezayi states with $k = 3$ and $M = 1$, i.e., $\nu = 3/5$, two distinct configurations are then expected, namely 1110011100 ··· and 1101011010 ··· (expressed in the occupation number representation),

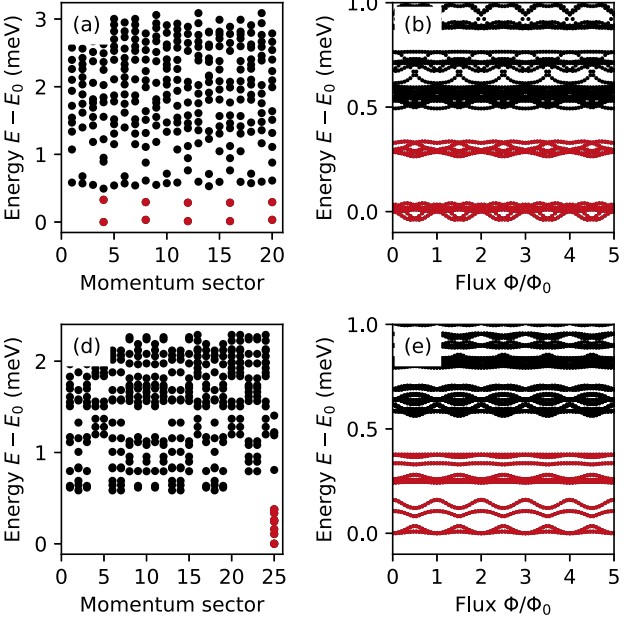

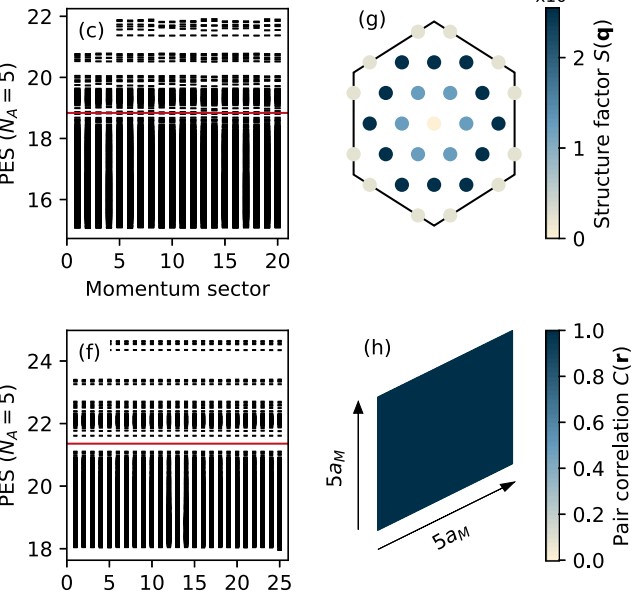

**Fig. 1 | Evidence for moiré parafermions at $\nu = 3/5$. a** Low-lying many-body energy spectrum, (**b**) spectral flow, and (**c**) $N_A = 5$ particle-cut entanglement spectrum for a $N_s = 20$-site system with $\gamma = 3.75$. The corresponding results for $N_s = 25$ sites are shown in (**d**–**f**). The 10-fold nearly-degenerate ground states are marked by red dots. In the PES, the number of states below the red solid line is 14404 for $N_s = 20$ and 51255 for $N_s = 25$, both matching the quasi-hole excitation counting for Read–

Rezayi states detailed in Supplementary Note 2. **g** Structure factor in the mini Brillouin zone and (**h**) pair-correlation function in the real-space unit cell for the Read–Rezayi ground states with $N_s = 25$. The spanning vectors for the considered systems are $\mathbf{T}_1 = (3, 4)$ and $\mathbf{T}_2 = (-2, 4)$ for $N_s = 20$, and $\mathbf{T}_1 = (5, 0)$ and $\mathbf{T}_2 = (0, 5)$ for $N_s = 25$.

which together with their translation-invariant partners result in a 10-fold degenerate ground state. Furthermore, the calculated ground states in Fig. 1(a), (d) appear at center-of-mass momenta matching those expected from the aforementioned thin-torus configurations.

While the approximate ground state degeneracy and the spectral flow are consistent with the $\mathbb{Z}_3$ RR phase, they could also correspond to weakly entangled states such as CDWs—as has been shown for candidate MR states[47]. Thus, more information is needed to conclusively determine whether the ground states in Fig. 1a, d correspond to RR states. An additional aspect pointing towards the RR phase is the many-body Chern number, which we calculate to be 3/5 for each ground state on average. We take one step further and obtain unambiguous evidence by computing the PES of the ground states[23,63]. The PES, which is distinct from the entanglement entropy and provides much richer information, is obtained by dividing the many-body system into $A$ and $B$ subsystems consisting of $N_A$ and $N_B = N_e - N_A$ particles, and then calculating the eigenvalues of $-\log \rho_A$, where $\rho_A = \text{tr}_B[\frac{1}{N_d} \sum_{i=1}^{N_d} |\Psi_i\rangle\langle\Psi_i|]$ is the reduced density matrix of $A$. Here, $\rho_A$ carries crucial information about the quasi-hole excitations in the $N_d$-fold degenerate ground states $|\Psi_i\rangle$, which are characteristically distinct for Abelian FCIs, non-Abelian FCIs, and CDWs. In particular, a gap in the PES is expected, below which the number of states exactly matches the amount of quasi-hole excitations allowed by the specific quantum phase of the system.

In Fig. 1c, f we show the PES with $N_A = 5$ for the two considered system sizes. In addition, we mark with a red line the entanglement energy below which the number of states matches exactly the amount of allowed quasi-hole excitations (the analytical counting is detailed in Supplementary Note 2). For the smaller system ($N_s = 20$), the dense low-energy sector is followed by a region of sparsely distributed states resembling a gap (around the red line in Fig. 1c), below which the state counting is similar to that expected for the RR phase. Remarkably, the PES for the larger system ($N_s = 25$) shows a clear entanglement gap with the low-energy state counting being exactly equal to the number of quasi-hole excitations in the RR phase. This observation serves as smoking-gun evidence that the considered system is in the $\mathbb{Z}_3$ RR phase.

To further clarify the nature of the phase, we calculate the structure factor $S(\mathbf{q})$ and pair-correlation function $C(\mathbf{r})$ averaged over the tenfold quasi-degenerate ground states, which provide insights on the crystalline or liquid character of the system (see Methods). For liquid phases, $C(\mathbf{r})$ approaches a constant for large $\mathbf{r}$, while it shows a periodic arrangement of charges for CDWs. As shown in Fig. 1g, h, both the structure factor and the pair-correlation function indicate a liquid behavior—concretely, no prominent symmetry-breaking peaks appear in $S(\mathbf{q})$ and $C(\mathbf{r})$ is constant.

## Stability of parafermion FCIs and quantum geometry

Having demonstrated the existence of non-Abelian RR states, we now explore the stability of such topological phases in the parameter space associated with the quantum geometry, which serves as a heuristic indicator of the analogy between the partially filled moiré band and the LLs in two-dimensional electron gases[49,64–68], although we note again that an accurate correspondence generally demands additional conditions[54]. For the $n$-th LL, the quantum metric and the Berry curvature satisfy[57] $\text{tr}[g(\mathbf{k})] = (2n + 1)|\Omega(\mathbf{k})|$, with $n = 0, 1, 2...$. Motivated by this property of LLs, it has been recently shown that both the Laughlin-like zero-energy modes[10] and non-Abelian MR state can be realized in a moiré flat band with a (nearly) ideal quantum geometry[42–46,48]. As mentioned before, the average quantum metric $\chi$ of the considered flat band in dTBG increases gradually from 1 to 3 as $\gamma$ is increased from 0 to infinity (see blue line in Fig. 2(a)). The parameter $\gamma$ then provides an ideal tuning knob to study the transition between the physics of bands with quantum geometric properties similar to the lowest LL and the second LL.

In the strong coupling limit, $\gamma \to \infty$, the targeted band acquires an ideal average quantum metric $\chi = 3$ and the band dispersion becomes

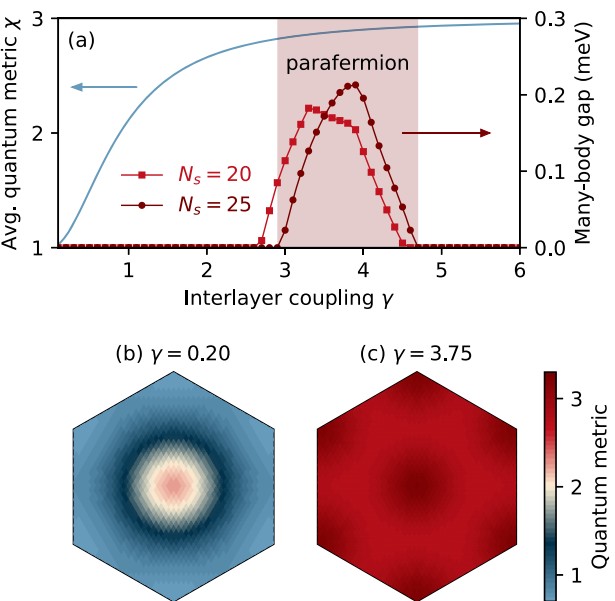

**Fig. 2 | Stability of the parafermion phase and quantum geometry. a** The many-body energy spectrum gap for non-Abelian $\mathbb{Z}_3$ RR states at $\nu = 3/5$ (red dotted lines, right axis) as a function of the coupling strength $\gamma$ between twisted layers is shown for system sizes $N_s = 20$ (squares) and $N_s = 25$ (circles) together with the averaged quantum metric $\chi$ (blue line, left axis). The energy gap is defined as the difference between the energies of the lowest non-RR state and the highest RR state (typically the 11th and the 10th lowest energies), and it is set to zero if the difference is negative. The shaded area denotes the regime where parafermions are stable. **b** Quantum metric $g(\mathbf{k})A_{BZ}/2\pi$ of the single-particle band for $\gamma = 0.20$ and (**c**) $\gamma = 3.75$. $A_{BZ}$ is the Brillouin zone area.

exactly flat as in the second LL. However, while the RR states are energetically competitive in the first excited Landau level[58,59], the many-body low-lying energy spectrum gap shown in Fig. 2a indicates their absence as $\gamma \to \infty$. Instead, the $\mathbb{Z}_3$ RR phase appears at an intermediate coupling strength, ranging from $\gamma \approx 3$ to 4.5. We note that the fluctuation of the quantum metric is minimized around this range of $\gamma$ (cf. Fig. 2b, c) and increases slightly for larger $\gamma$ values (see Supplementary Fig. 2). These findings strongly suggest that the quantum geometry criterion is not sufficient to diagnose the non-Abelian topological phases. It nevertheless gives some intuition of where exotic states may occur—the RR states do in fact form in a region where the quantum geometry resembles that of the first excited rather than that of the lowest LL. We note that there are alternative ways of making moiré minibands that are close but not identical to the first excited LL such as considering other magic angles[69], adding a "negative" magnetic field[70] or periodic strain[71], and considering bands further away from charge neutrality[42,43,46,48].

In the weak coupling limit $\gamma \to 0$, where the system approaches to two decoupled chiral TBG sheets, the average quantum metric $\chi \to 1$ shown in Fig. 2a and the quantum geometry distribution shown in Fig. 2b and Supplementary Fig. 2 suggest that the targeted flat band resembles the lowest LL, where Abelian hierarchy states at band filling $\nu = 3/5$ and its particle-hole partner $\nu = 2/5$[35–37] are indeed present (see Supplementary Note 3).

## Particle-hole asymmetry and hole entanglement spectrum

We now search for the possibility of RR states at electronic band filling $\nu = 2/5$, which in ideal LLs correspond to the particle-hole conjugate of the $\nu = 3/5$ RR phase. It has been shown that, in general, particle-hole symmetry is broken in FCIs and moiré systems[8,72]—a fact that can be seen as a consequence of a fluctuating quantum metric[73,74]. In line with this asymmetry, we do not find a clear many-body energy gap of the RR

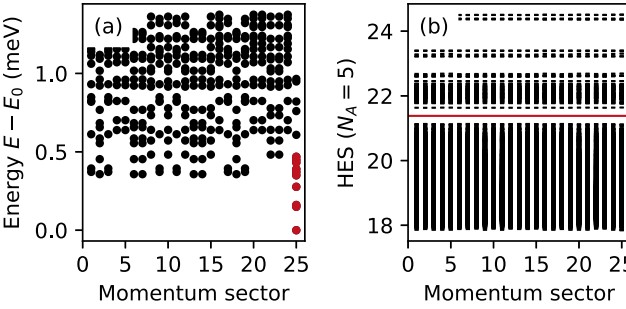

**Fig. 3 | Moiré parafermions and competing states at $\nu$ = 2/5. a** Low-lying energy spectrum at band filling $\nu$ = 2/5 and (**b**) the respective hole entanglement spectrum (HES) for $\gamma$ = 3.75 in a system with size $N_s$ = 25. In the HES, the number of states below the first entanglement gap (red line) is 51255, matching the quasi-particle counting of RR states at hole-filling 3/5.

states for filling $\nu$ = 2/5 at the system sizes considered, suggesting that the RR phase is less stable for $\nu$ = 2/5 than for $\nu$ = 3/5 (see Fig. 3a). In addition, although the tenfold quasi-degenerate states are still present at the correct momentum sector, their corresponding PES does not show a gap with the correct quasi-hole counting (see Supplementary Note 4 and Supplementary Fig. 5). This, however, should not be surprising: since the $\nu$ = 2/5 RR phase can be understood as the RR phase for holes at 3/5 hole-filling, adding holes to the system directly increases the ground-state energy as the 4-body interaction can no longer be minimized.

Instead, the $\nu$ = 2/5 RR phase allows a certain number of quasi-particle excitations without increasing the ground-state energy. This is reflected in the hole entanglement spectrum (HES), where the many-body system is divided into $A$ and $B$ subsystems with $N_A$ and $N_B = N_s - N_e - N_A$ now denoting the number of holes. Importantly, the HES (Fig. 3b) displays an entanglement gap below which the number of states matches the quasi-particle counting of the 3/5 hole-filling (i.e. $\nu$ = 2/5) RR phase, indicating that the calculated states indeed correspond to the non-Abelian RR phase. From a topological perspective, this then implies that the nature of the RR states remains largely unaffected by the renormalization of the hole dispersion that is expected to arise from fluctuations in the quantum metric[72], even though these fluctuations may alter the ground state energies as suggested by the absence of a gap in Fig. 3a.

## Discussion

In this work we have demonstrated the emergence of Fibonacci ($\mathbb{Z}_3$) parafermions in an exemplary moiré system based on dTBG. While a quantitatively realistic material prediction would require a consideration of finite bandwidth[75] and band mixing effects[76–78] as well as reliable estimates of system parameters (which in many cases remain controversial), our results are very encouraging for several reasons.

First, that an unscreened Coulomb interaction is sufficient to obtain parafermions is quite remarkable given that, for the simplest non-Abelian FCIs, fine-tuned interactions are necessary even in toy models[79,80].

Second, although there is suggestive evidence for the possibility of RR parafermion quantum Hall states in Landau levels at $\nu$ = 12/5 and 13/5[58,59], competing orders make the case less compelling than that of MR states at $\nu$ = 5/2. This situation may be opposite in the moiré context: at half filling of moiré bands charge density waves[15] and the anomalous Hall crystal[81] are additional fierce competitors with no direct LL analogues, while presumably no similarly competitive non-LL states exist at $\nu$ = 3/5 due to a lack of natural ordering patterns.

Third, bands with an average quantum geometry equal to that of excited Landau levels are not necessarily optimal for realizing RR parafermions. Figure 2 instead suggests that the most optimal bands

have an average quantum geometry intermediate between the lowest and first excited Landau levels. It is quite likely that the optimal range of $\chi$ depends on the model/material and is related to the band flatness and quantum metric fluctuations. In this sense, the tunability of moiré bands offers enormous potential to achieve bands with the proper quantum geometric properties.

Fourth, although we find the $\mathbb{Z}_3$ RR parafermion states to be energetically favoured in a moderate parameter range (cf. Fig. 2a), it is striking that, even in cases when it is not apparent from the small system sizes avaliable whether the RR states will be competitive for a macroscopic number of particles, the entanglement spectrum shows telltale signatures of the RR states (cf. Fig. 3) suggesting that these states may prevail in a larger parameter range.

Finally, we emphasize that Fibonacci parafermions are arguably more desirable and elusive than Majorana fermions for fundamental reasons[82]: unlike the Majoranas, they are not possible to realize in effectively non-interacting Hermitian models[53] and they potentially serve as building blocks for universal quantum computing[82]. These aspects have motivated the study of highly complex and technically extremely challenging setups such as FQH–superconductor heterostructures[51]. In contrast, moiré heterostructures offer a simple, versatile and tunable platform which does in principle not even require an external magnetic field.

In addition, we note that although the expected Hall conductivity for the RR phase is the same as for the Abelian hierarchy state at equal filling factor, many techniques proposed for quantum Hall systems could be imported to moiré systems in order to experimentally discern these two phases. In particular, the braiding statistics of parafermions can be probed via two-point contact interferometry[83–85]. Moreover, frequency-dependent shot noise measurements in Mach-Zehnder interferometers could provide access not only to the fractional charge (which is equal in the Abelian and $\mathbb{Z}_3$ RR parafermion phases) but also to the non-Abelian nature of parafermions[86]. Other proposals exploit non-linear $I - V$ characteristics in tunneling experiments[87] or Coulomb blockade in quantum dots[88,89].

We hope that these observations will serve as inspiration for both the theoretical and experimental pursuit of parafermions emerging in moiré minibands.

## Methods
### Quantum metric of the single-particle band
The quantum geometric properties of the target single-particle band are determined by $\phi(\mathbf{k})$, i.e. the corresponding eigenstate of the momentum-space Hamiltonian, $H_0(\mathbf{k})$. Here, $H_0(\mathbf{k})$ is a matrix in the space of layers, sublattices, and the subbands that arise from zone-folding $\mathbf{k}$-points outside of the moiré Brillouin zone up to a certain cutoff $N_{cutoff}$. Each subband element is determined by moiré reciprocal lattice vectors $\mathbf{G}$ and $\mathbf{G}'$ and corresponds to $H_0(\mathbf{k})|_{\mathbf{GG}'} = \int d^2\mathbf{r}\, e^{-i(\mathbf{k}+\mathbf{G})\cdot\mathbf{r}} H_0(\mathbf{r}) e^{i(\mathbf{k}+\mathbf{G}')\cdot\mathbf{r}}$. The quantum (Fubini–Study) metric $g_{ij}(\mathbf{k})$ describes the distance (or form factors) between states with a small momentum difference. Concretely, it is the real part of the quantum geometric tensor $\mathcal{Q}_{ij}(\mathbf{k})$, i.e. $g_{ij}(\mathbf{k}) = \mathrm{Re}[\mathcal{Q}_{ij}(\mathbf{k})]$ where $\mathcal{Q}_{ij}(\mathbf{k}) = \partial_i\phi^\dagger(\mathbf{k})\partial_j\phi(\mathbf{k}) - [\partial_i\phi^\dagger(\mathbf{k})\phi(\mathbf{k})][\phi^\dagger(\mathbf{k})\partial_j\phi(\mathbf{k})]$ $\qquad (\partial_i \equiv \partial_{k_i}$ with $i = x, y)$.

### Exact diagonalization
In order to tackle the many-body problem of interacting electrons, we first diagonalize $H_0(\mathbf{r})$ in the basis of Bloch states and then project the electron-electron interactions onto the considered nearly-flat band, obtaining $H_{int} = \frac{1}{2}\sum_{\mathbf{k_1k_2k_3k_4}} V_{\mathbf{k_1k_2k_3k_4}} c^\dagger_{\mathbf{k_1}} c^\dagger_{\mathbf{k_2}} c_{\mathbf{k_3}} c_{\mathbf{k_4}}$. Here, the matrix element $V_{\mathbf{k_1k_2k_3k_4}}$ ensures momentum conservation and contains single-particle form factors[8,90] as well as the 2D bare Coulomb potential $V(\mathbf{q}) = \frac{e_0^2}{2A\epsilon\epsilon_0|\mathbf{q}|}$ with the average dielectric constant of the system $\epsilon = 5$[91]. Next, we obtain the low-lying energy spectrum of many-body states by exact diagonalization of the spin- and valley-polarized interaction

Hamiltonian $H_{int}$. In particular, we consider a system of $N_e$ electrons in $N_s$ moiré sites with the filling $\nu = N_e/N_s = 3/5, 2/5$. We ensure that the calculations are converged with respect to the number of moiré sub-bands, $(2N_{cutoff} + 1)^2$, and note that $N_{cutoff} = 4$ is sufficient. In addition, we note that calculations in finite-size systems yield results that deviate with respect to infinite systems due to finite size effects. Here, we have employed state of the art methods and considered systems as large as possible to minimize such uncertainties.

### Structure factor and pair correlation

The structure factor helps measure if the ground states exhibit a preferred charge order and is defined as $S(\mathbf{q}) = \frac{1}{N_e^2}\langle \rho(\mathbf{q})\rho(-\mathbf{q})\rangle - \delta_{\mathbf{q},0}$, with $\rho(\mathbf{q})$ being the density operator projected onto the targeted nearly flat band. On the other hand, the real space pair-correlation function reads $C(\mathbf{r}) = \frac{1}{N_e^2}\langle n(\mathbf{r})n(\mathbf{0})\rangle$, where $n(\mathbf{r})$ is the real-space density operator. CDW order that breaks the translation symmetry of the underlying moiré lattice is characterized by structure factor peaks at symmetry points of the Brillouin zone, whereas liquids are distinguished by the absence of such peaks. Symmetry-breaking CDW peaks in $S(\mathbf{q})$ would correspond to a periodic modulation of the pair-correlation function with a periodicity different than that of the underlying moiré lattice.

## Data availability
The data generated in this study and presented in the article are provided in the Source Data file. Source data are provided with this paper.

## Code availability
The codes used to generate and analyse the data are available from the corresponding author upon request.

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

## Acknowledgements

We acknowledge useful discussions and related collaborations with Ahmed Abouelkomsan, Liang Fu, Zhao Liu, Aidan Reddy and Donna Sheng. This work was supported by the Swedish Research Council (VR, grant 2018-00313), the Wallenberg Academy Fellows program of the Knut and Alice Wallenberg Foundation (2018.0460) and the Göran Gustafsson Foundation for Research in Natural Sciences and Medicine. The computations were enabled by resources provided by the National Academic Infrastructure for Supercomputing in Sweden (NAISS), partially funded by the Swedish Research Council through grant agreement no. 2022-06725.

## Author contributions

E.B. conceived the research. H.L. and R.P.-C. performed the numerical calculations. All authors analyzed and discussed the results and wrote the manuscript.

## Funding

## Competing interests

The authors declare no competing interests.
