## [Transparent Peer Review file · Nature Communications]

Parafermions in moiré minibands

Corresponding Author: Dr Raul Perea-Causin

Version 0:

Reviewer comments:

Reviewer #1

(Remarks to the Author)

The manuscript, "Parafermions in Moiré Minibands," explores the theoretical realization of Fibonacci parafermions within moiré minibands, focusing specifically on double twisted bilayer graphene systems. By employing exact diagonalization, the study provides numerical evidence for a non-Abelian fractional Chern insulator (FCI) phase, namely the Z3 Read-Rezayi (RR) phase. These findings suggest a promising pathway for utilizing moiré materials in topological quantum computation, where the unique properties of Fibonacci parafermions could offer significant advantages over other quasiparticles, such as Majorana fermions.

The recent experimental observation of FCI states in moiré superlattices has generated enormous excitement in the field. A key research question now is to identify even more exotic FCI states in moiré superlattices analogous to the non-abelian fractional quantum Hall states, or potentially surpassing the known states within fractional quantum Hall systems. This manuscript contributes to this ongoing investigation, presenting compelling evidence for the RR state through analyses of ground state degeneracy, flux insertion, and entanglement entropy. While the model presented is somewhat idealized due to the chiral limit assumption (which is nonetheless motivated by the model Hamiltonian for twisted double bilayer graphene), the work paves the way for further theoretical efforts toward identifying a more realistic model that supports the RR state.

The manuscript is well-written, with clear presentation; however, one crucial point requires clarification before a publication decision can be made. Although the n th Landau level possesses the convenient property that the average quantum metric $\chi = 2n + 1$, the reverse implication is not necessarily valid—i.e., $\chi = 3$ does not imply that the Chern band resembles the second Landau level. This could be due to the mixing of different Landau levels, which also connects to questions about the conditions necessary to realize the RR state. Quantum metric is certainly an important quantity for characterizing a topological flat band, but does the stabilization of the RR state require χ to lie within the range specified in Fig. 2, or is it model-dependent? This point warrants clarification from the authors.

The authors may also want to move some results from Supplement to the main text. For instance, the results in Fig. S2 are informative to characterize the single-particle topological band.

Finally, the authors should discuss how the RR state might be detected experimentally and how it could be distinguished from an Abelian FCI state at the same filling.

Reviewer #2

(Remarks to the Author)

Reviewer #3

(Remarks to the Author)

The authors provide numerical evidence (based on exact diagonalization) for the existence of the Z3 Read-Rezayi parafermion phase in a moire system.

The paper is scientifically sound, interesting and timely, and thus I recommend publication. Here are some specific observations:

In the intro, "conclusive evidence" sounds like an oversell. I recommend talking about "numeric evidence" instead.

In the second paragraph of page 3, the expression for entanglement entropy is missing the first density matrix. It should be $-\rho \log \rho$, not just $-\log \rho$.

The explanation of the two distinct configurations of the state in the occupation number representation is not clear to me. The sequences are 1110011100... and 1101011010... however, this is a 2D system. What do you mean by "consecutive sites"?

On the third point of the discussion section, I don't get what the authors mean by 'mimicking' Landau levels (LLs). Please elaborate.

Version 1:

Reviewer comments:

Reviewer #1

(Remarks to the Author)

The authors have satisfactorily addressed the comments and revised the manuscript accordingly. I recommend the current version of the manuscript for publication.

Reviewer #2

(Remarks to the Author)

Response to the reviewers' comments

We thank the reviewers for carefully reading our manuscript and for providing valuable feedback which helped us to further improve the clarity of our manuscript. We provide a detailed point-by-point response to the comments below. Changes in the revised manuscript are marked in blue.

Reviewer: 1

- 1. Comment** “The manuscript, ‘Parafermions in Moiré Minibands,’ explores the theoretical realization of Fibonacci parafermions within moiré minibands, focusing specifically on double twisted bilayer graphene systems. By employing exact diagonalization, the study provides numerical evidence for a non-Abelian fractional Chern insulator (FCI) phase, namely the Z_3 Read-Rezayi (RR) phase. These findings suggest a promising pathway for utilizing moiré materials in topological quantum computation, where the unique properties of Fibonacci parafermions could offer significant advantages over other quasiparticles, such as Majorana fermions. The recent experimental observation of FCI states in moiré superlattices has generated enormous excitement in the field. A key research question now is to identify even more exotic FCI states in moiré superlattices analogous to the non-abelian fractional quantum Hall states, or potentially surpassing the known states within fractional quantum Hall systems. This manuscript contributes to this ongoing investigation, presenting compelling evidence for the RR state through analyses of ground state degeneracy, flux insertion, and entanglement entropy. While the model presented is somewhat idealized due to the chiral limit assumption (which is nonetheless motivated by the model Hamiltonian for twisted double bilayer graphene), the work paves the way for further theoretical efforts toward identifying a more realistic model that supports the RR state. The manuscript is well-written, with clear presentation; however, one crucial point requires clarification before a publication decision can be made.”

Answer We thank the referee for the clear assessment of our work, and in particular for pointing out the relevance of our results for potentially utilizing moiré materials in topological quantum computing.

- 2. Comment** “Although the n th Landau level possesses the convenient property that the average quantum metric $\chi = 2n + 1$, the reverse implication is not necessarily valid—i.e., $\chi = 3$ does not imply that the Chern band resembles the second Landau level. This could be due to the mixing of different Landau levels, which also connects to questions about the conditions necessary to realize the RR state. Quantum metric is certainly an important quantity for characterizing a topological flat band, but does the stabilization of the RR state require χ to lie within the range specified in Fig. 2, or is it model-dependent? This point warrants clarification from the authors.”

Answer This is indeed an important point. We agree that $\chi = 3$ by itself does not imply that the band is equivalent to the first excited (second) Landau level, as discussed for example in arXiv:2403.00856. We have now modified the language used in the manuscript to avoid confusion on this aspect.

Regarding the referee’s question, we suspect that the optimal range of χ is model dependent and is related to the band flatness and quantum metric fluctuations. We would like to emphasize that the main point of our work is to show the potential of moiré materials for hosting Fibonacci parafermions. While we attempt to provide some insights based on quantum geometry into what makes this phase stable, further studies

(which we hope will be motivated by our work) are needed to obtain a more complete picture. In any case, the question raised by the referee is valuable and deserves to be discussed in the manuscript; therefore we have added the following sentences in the discussion part:

“It is quite likely that the optimal range of χ depends on the model/material and is related to the band flatness and quantum metric fluctuations. In this sense, the tunability of moiré bands offers enormous potential to achieve bands with the proper quantum geometric properties.”

- 3. Comment** “The authors may also want to move some results from Supplement to the main text. For instance, the results in Fig. S2 are informative to characterize the single-particle topological band.”

Answer We appreciate this suggestion and have accordingly moved the plots of quantum metric distribution (Fig. S2(c)-(d)) to the main text (Fig. 2).

- 4. Comment** “Finally, the authors should discuss how the RR state might be detected experimentally and how it could be distinguished from an Abelian FCI state at the same filling.”

Answer We thank the referee for this valuable suggestion. We have added the following paragraph in the discussion section:

“In addition, we note that although the expected Hall conductivity for the RR phase is the same as for the Abelian hierarchy state at equal filling factor, many techniques proposed for quantum Hall systems could be imported to moiré systems in order to experimentally discern these two phases. In particular, the braiding statistics of parafermions can be probed via two-point contact interferometry [PRL 96, 016803 (2006), Nat. Phys. 16, 931–936 (2020), PRX 13, 041012 (2023)]. Moreover, frequency-dependent shot noise measurements in Mach-Zehnder interferometers could provide access not only to the fractional charge (which is equal in the Abelian and \mathbb{Z}_3 RR parafermion phases) but also to the non-Abelian nature of parafermions [PRB 76, 085333 (2007)]. Other proposals exploit non-linear $I - V$ characteristics in tunneling experiments [PRB 61, 5473 (2000)] or Coulomb blockade in quantum dots [PRL 100, 086803 (2008), Nucl. Phys. B 899, 289-311 (2015)].”

Reviewer: 3

- 1. Comment** “The authors provide numerical evidence (based on exact diagonalization) for the existence of the \mathbb{Z}_3 Read-Rezayi parafermion phase in a moire system. The paper is scientifically sound, interesting and timely, and thus I recommend publication. Here are some specific observations:”

Answer We thank the referee for their clear recommendation for publication of our work and for their constructive feedback.

- 2. Comment** “In the intro, ‘conclusive evidence’ sounds like an oversell. I recommend talking about ‘numeric evidence’ instead.”

Answer We have followed the referee’s suggestion and changed the text to “numerical evidence”.

3. Comment “In the second paragraph of page 3, the expression for entanglement entropy is missing the first density matrix. It should be $-\rho \log \rho$, not just $-\log \rho$.”

Answer We would like to clarify that the quantity that we calculate is not the entanglement entropy but rather the entanglement spectrum introduced in PRL 101, 010504 (2008); in particular we consider the particle partition discussed in PRL 106, 100405 (2011). The entanglement spectrum consists of the eigenvalues of a (Hermitian) Hamiltonian-like operator \tilde{H} that emerges by conveniently defining the reduced density matrix as $\rho = \exp(-\tilde{H})$. The whole spectrum of eigenvalues provides more complete information than the entanglement entropy, which is just a single number. In order to make this distinction clear, we have added the following text: “The PES, which is distinct than the entanglement entropy and provides much richer information, is obtained by [...]”.

4. Comment “The explanation of the two distinct configurations of the state in the occupation number representation is not clear to me. The sequences are 1110011100... and 1101011010... however, this is a 2D system. What do you mean by ‘consecutive sites’?”

Answer We thank the referee for pointing out that this important aspect needs further clarification. The sites we refer to appear in the context of the thin-torus limit, in which one can analytically calculate the many-body ground state degeneracy of fractional quantum Hall states; see e.g. PRL 94, 026802 (2005). In particular, a periodic quantum Hall system (equivalent to a torus) can be mapped into a 1D lattice e.g. in the y-direction, where the lattice sites are directly related to the discrete momenta in the perpendicular x-direction. In the thin-torus limit, where the x-direction size of the periodic unit cell is small, the problem of interacting electrons can be solved analytically. Applying this approach to the Read-Rezayi phase that we discuss, the ten degenerate ground states are seen to consist of the sequences 1110011100... and 1101011010... (as well as translation invariant partners) in the 1D lattice, which in this context is analogous to the discrete momentum space. Since the thin-torus limit is adiabatically connected to the 2D quantum Hall system, the tenfold ground state degeneracy persists in the latter and is one of the fingerprints that allow the identification of the Read-Rezayi phase in numerical calculations.

We have extended the discussion on the thin torus as follows:

“The 10-fold ground state degeneracy is in agreement with that expected in the thin-torus limit, where a quantum Hall system is adiabatically mapped into a 1D lattice that can be exactly solved. In particular, the RR ground states at filling $\nu = k/(kM + 2)$ are formed by k particles occupying $kM + 2$ consecutive sites and every two particles being separated by at least M . Note that the 1D lattice sites in the thin torus are analogous to momentum points.”

5. Comment “On the third point of the discussion section, I don’t get what the authors mean by ‘mimicking’ Landau levels (LLs). Please elaborate.”

Answer Here we are referring to bands that have properties similar to Landau levels. Concretely, what we intend to say is that our results indicate that bands with quantum geometric properties similar to those of excited Landau levels are not necessarily optimal for realizing non-Abelian phases. We have rephrased the sentence as follows to make it more clear:

“Third, bands with an average quantum geometry similar to that of excited Landau levels are not necessarily optimal for realizing RR parafermions.”